# Mixture of Experts for Image Classification: What's the Sweet Spot?

**Mathurin Videau**                                    *mvideau@meta.com*
*Meta AI, TAU, INRIA, and LISN (CNRS & Univ. Paris-Saclay)*

**Alessandro Leite**                                   *aleite@insa-rouen.fr*
*INSA Rouen Normandy, University of Rouen Normandy, LITIS UR 4108*

**Marc Schoenauer**                                    *marc.schoenauer@inria.fr*
*TAU, INRIA and LISN (CNRS & Univ. Paris-Saclay)*

**Olivier Teytaud**
*Thales, CortAIx-Labs*

**Reviewed on OpenReview:** *https://openreview.net/forum?id=hKise4AJgp*

## Abstract

Mixture-of-Experts (MoE) models have shown promising potential for parameter-efficient scaling across domains. However, their application to image classification remains limited, often requiring billion-scale datasets to be competitive. In this work, we explore the integration of MoE layers into image classification architectures using open datasets. We conduct a systematic analysis across different MoE configurations and model scales. We find that moderate parameter activation per sample provides the best trade-off between performance and efficiency. However, as the number of activated parameters increases, the benefits of MoE diminish. Our analysis yields several practical insights for vision MoE design. First, MoE layers most effectively strengthen tiny and mid-sized models, while gains taper off for large-capacity networks and do not redefine state-of-the-art ImageNet performance. Second, a *Last-2* placement heuristic offers the most robust cross-architecture choice, with *Every-2* slightly better for Vision Transform (ViT), and both remaining effective as data and model scale increase. Third, larger datasets (e.g., ImageNet-21k) allow more experts, up to 16, for ConvNeXt to be utilized effectively without changing placement, as increased data reduces overfitting and promotes broader expert specialization. Finally, a simple linear router performs best, suggesting that additional routing complexity yields no consistent benefit.

## 1 Introduction

Recent advances in machine learning, particularly in the domains of natural language processing (NLP)(Vaswani et al., 2017; Kenton & Toutanova, 2019) and computer vision (Dosovitskiy et al., 2020), have been primarily driven by scaling up model size, computational budgets, and training data. Although these large-scale models demonstrate impressive performance, they are often expensive to train and consume considerable energy resources (Strubell et al., 2019). As a result, the research community has become increasingly interested in exploring more efficient training and serving paradigms. One such promising solution is the use of sparse expert models, with Mixture-of-Experts (MoE) (Shazeer et al., 2017; Lepikhin et al., 2020) emerging as a popular variant.

MoE models introduce sparsity by partitioning the set of parameters into multiple parallel sub-models, called "experts." During training and inference, the gating part of the models routes input examples to specific expert(s), ensuring that each example only interacts with a subset of the network parameters. As the

computational cost is partially correlated with the number of parameters activated for a given sample rather than the total number of parameters, this approach facilitates the scaling-up of the model, while keeping computational costs under control, making it an attractive option for a wide range of applications.

Despite their success in various domains (Hihn & Braun, 2021; Costa-jussà et al., 2022; Zoph et al., 2022; Fedus et al., 2022), the application of MoE models in image classification and computer vision, in general, remains limited, and it often requires very large datasets (Riquelme et al., 2021; Mustafa et al., 2022; Komatsuzaki et al., 2022) to be competitive against state-of-the-art approaches. In this work, we focus on leveraging the potential of MoE models for image classification on ImageNet-1k and ImageNet-21k (Russakovsky et al., 2015). We study the efficiency of integrating MoE within two renowned architectures, ConvNext (Liu et al., 2022) and Vision Transformer (ViT) (Touvron et al., 2022). We conduct a series of experiments considering various architecture configurations. Likewise, we investigate the impact of various components, including the number of experts and their sizes, the gate design, and the layer positions, among others. Our experimental findings indicate that optimal design is contingent on the specific network architecture, and that consistently situating the MoE layer within the final two even blocks invariably yields substantial improvements for moderate model size. Nevertheless, when scaling up the approach to large models and datasets close to the state-of-the-art, the benefits of using Mixture-of-Experts for image classification gradually vanish.

**Contributions.** In this work, we provide a systematic study of applying Mixture-of-Experts (MoE) models to image classification tasks, focusing on the ImageNet-1k and ImageNet-21kt benchmarks. Training follows supervised image classification in two setups, either on ImageNet-1K or ImageNet-22K. Unlike prior efforts that emphasize large-scale deployments with vast compute resources, we explore MoE integration within mid-sized ConvNeXt and Vision Transformer (ViT) architectures, identifying effective design principles for efficient training and inference. Our analysis yields the following practical guidance for vision MoE design:

- **When MoE helps (scale):** MoE strengthens tiny and mid-sized models, but gains diminish for large-capacity networks, and do not redefine state-of-the-art ImageNet performance.

- **Placement heuristic:** A *Last-2* configuration is the most robust cross-architecture choice, with *Every-2* slightly better for ViT. These placements remain effective when scaling data or model size.

- **Scaling with data:** With larger datasets (e.g., ImageNet-21k), more experts can be used effectively, up to 16 for ConvNeXt, without changing placement. Increased data reduces overfitting and enables broader expert utilization.

- **Routing choice:** A simple linear router works the best, indicating that added routing complexity brings no consistent benefit.

## 2 Related Work

The Mixture-of-Experts (MoE) model, introduced by (Jacobs et al., 1991), partitions complex tasks into hopefully simpler sub-tasks handled by expert models, whose predictions are combined to produce the final output. This framework has been successfully integrated into various neural network architectures, particularly transformers for NLP tasks (Shazeer et al., 2017). Given this success in NLP, interest has increased in applying MoE to the diverse and complex computer vision tasks.

The transformative potential of MoE architectures is underscored by their successful integration across various domains, with the Vision Transformer (ViT) being a prime example. Riquelme et al. (2021) introduced V-MoE, an MoE-augmented ViT model, for image classification tasks on a massive dataset containing hundreds of millions of examples. They showed that V-MoE not only matches the performance of prior state-of-the-art architectures but also requires half the computational resources during inference. Lou et al. (2021) presented a sparse MoE MLP model based on the MLP-Mixer architecture (Tolstikhin et al., 2021). While the MLP-Mixer architecture does not display performance accuracy, the MoE-enhanced version surpassed its dense counterpart in experiments on ImageNet and CIFAR. Hwang et al. (2023a) demonstrated the effectiveness of SwinV2-MoE, a MoE-based model built upon Swin Transformer V2 architecture. They reported superior accuracy in

downstream computer vision tasks. Similarly, Puigcerver et al. (2024) introduced a soft MoE mechanism, demonstrating improved performance and training speed compared to classical MoE on billion-scale datasets. More recently, Han et al. (2024) proposed ViMoE, which investigates expert placement strategies by fine-tuning DINOv2 models. In contrast, our work focuses on training MoE architectures from scratch, enabling a systematic study of their behavior across both ConvNeXt and ViT backbones. This distinction allows us to provide complementary insights into how MoE mechanisms contribute to compute efficiency and performance in vision models.

## 3   Sparsely Activated MoE

Based on conditional computations, MoE aims at activating specific parts of the model depending on the input. The core idea is to assign experts to different regions of the input space, thereby increasing the model capacity by augmenting the number of parameters without incurring significant computational overhead (Jacobs et al., 1991).

According to Shazeer et al. (2017), an MoE includes a router $G$ and a set of experts $E$. The router learns a sparse assignment between the input and the experts, while the experts process the inputs like standard neural network modules do. Let $x$ be the input, and let $(E_i)_{i \in [1,N]}$ be the $N$ experts of the MoE layer. Following (Shazeer et al., 2017), the output of this MoE layer is given by:

$$\sum_{i \in \text{Top}_k(x)} G(x)_i \cdot E_i(x)$$

where, $G$ is a conv1x1 gate employing a softmax function, and $G(x)_i$ is its $i^{th}$ output. To promote sparsity in the MoE layer, the number of participating experts is restricted to $k < N$ for each input: $\text{Top}_k(x)$ contains the indices of the $k$ highest $G(x)_i$. Note that involving $k > 1$ experts increases the computational cost compared to the sparse ($k = 1$) counterpart. We tested several gate designs, but none of them clearly outperformed the simple conv1x1 gate (Sec. 4.3.5).

However, enforcing this MoE design is insufficient to yield an effective MoE. To exploit compute capacity optimally, we encourage uniform usage of each expert by using a load balancing auxiliary loss proposed in (Shazeer et al., 2017), which promotes balanced expert assignments and equal importance of experts (for more details, see Section 4 of Shazeer et al. (2017)). We also use Batch Prioritized Routing (BPR) introduced by (Riquelme et al., 2021).

### 3.1   Vision Transformer

The Vision Transformer (ViT) (Dosovitskiy et al., 2020) represents a paradigm shift in computer vision. It moves away from traditional convolution-based architectures in favor of the Transformer's attention mechanism, initially pioneered for natural language processing tasks (Vaswani et al., 2017). In the ViT model, an image is segmented into fixed-size and non-overlapping patches. These patches are linearly embedded into vectors and then passed through a series of Transformer blocks. Each Transformer block is predominantly made up of two main components: a multi-head self-attention mechanism and a multi-layer perceptron (MLP). Within the context of ViT, the MLP ratio (denoted mlp_ratio in the following) refers to the ratio of the hidden dimensions of the MLP to the embedding dimensions. This ratio is pivotal, as it determines the capacity and computational cost of the MLP component. We replace some predefined MLP block by a MoE, as depicted in Fig. 1a. Each MoE operates at the patch level: the experts are spatially distributed. We consider (a) a standard ViT architecture (ViT-B (Base)) and (b) a variant with 8 experts (ViT-B-8).

### 3.2   ConvNeXt

ConvNeXt (Liu et al., 2022) has modernized the classical convolutional neural networks (CNN) design by incorporating insights from Vision Transformers (ViT). The central component of the ConvNeXt architecture is a block containing a $7 \times 7$ depth-wise convolution followed by a $1 \times 1$ convolutional layer (Fig. 1b-left). This block is replicated across four stages, each with decreasing resolution. In this work, we replace some predefined

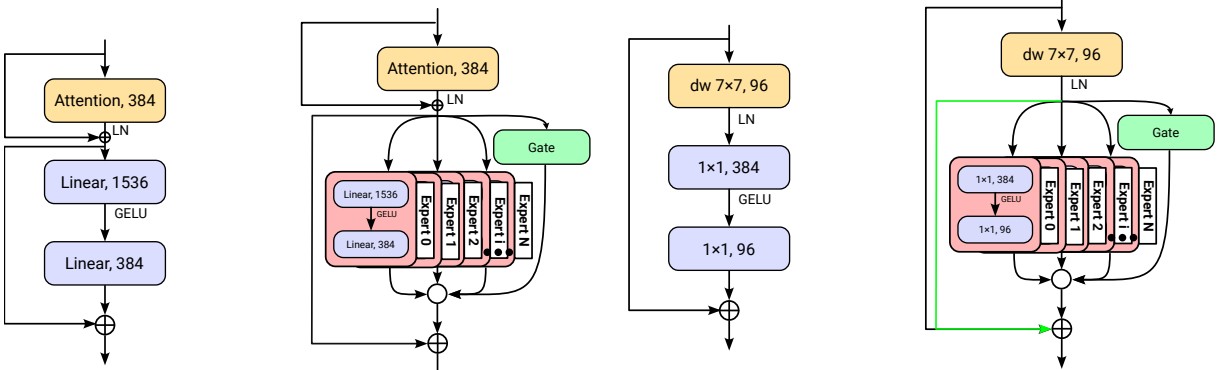

(a) **Left**: ViT base block. **Right**: Our modified MoE ViT block. LN: Layer normalization. GELU: Gaussian Error Linear Units.

(b) **Left**: ConvNeXt original base block. **Right**: The modified MoE ConvNeXt block. The green link represents the added skip connection. DW: Depth-wise. LN: Layer normalization.

Figure 1: Vision Transformer and ConvNext architectures.

ConvNeXt block, namely the multilayer perceptron, with a MoE block, and introduce a skip connection, as illustrated in Fig. 1b-right. Consequently, each expert becomes a fully convolutional block operating at the feature map level, resulting in spatially distributed experts. Hence, this design not only increases the network capacity but also allows each expert to specialize in specific spatial locations and engages multiple experts per image. We examine three variants: (a) ConvNeXt-T (Tiny), ConvNeXt-S (Small), and ConvNeXt-B (Base). These architectures primarily differ in depth and width; and (b) ConvNeXt-S-4 and ConvNeXt-B-4, which are variants with 4 experts.

## 4 Experiments on ImageNet

This section presents the results obtained by the architectures described above (that incorporate MoE into ConvNeXt and ViT architectures), trained on ImageNet-1k or pre-trained on ImageNet-21k datasets. It includes the experimental setup, the results on the ImageNet-1k validation set, and the impact of various MoE configurations.

### 4.1 Experimental Setup

All the results presented in this section rely on MoE models trained on the ImageNet datasets. Although the context is different due to the changes in network structures presented in Secs. 3.1 and 3.2, we set the training hyperparameters similar to (Touvron et al., 2022) for ViT and (Liu et al., 2022) for ConvNeXt (a grid search looking for better settings did not bring significant improvement).

Furthermore, when working with the ImageNet-1k dataset, we use a strong data-augmentation pipeline, including Mixup (Zhang et al., 2018), Cutmix (Yun et al., 2019), RandAugment (Cubuk et al., 2020), and Random Erasing (Zhong et al., 2020), over 300 epochs. Likewise, we utilize drop path, weight decay, and expert-specific weight decay as regularization strategies. However, for the pretraining phase on ImageNet-21k, we exclude Mixup and Random Erasing from the data augmentation pipeline because they did not improve the results, as also reported in (Touvron et al., 2022; Tu et al., 2022). We pre-train the model for 90 epochs, adhering to the original ConvNeXt approach (Liu et al., 2022) and, consistently with (Liu et al., 2022) and (Touvron et al., 2022), the final results of ImageNet-1k are obtained through fine-tuning for 30 epochs for ConvNeXt and 50 for ViT. Comprehensive details of all the hyperparameters are provided in Tab. 10 in App. A.

Table 1: Accuracy for different ImageNet-1K trained models and the MoE strategies "Every 2" and "Last 2" (see text). "Top $k$" corresponds to the number of experts involved. Throughput is measured on V100 GPUs, following (Touvron et al., 2021). For ImageNet-1k, non-isotropic ConvNeXt models feature an mlp_ratio of 2, which contributes to their improved flops-per-sample efficiency compared to their dense counterparts.

| Architecture | #Params $(\times 10^6)$ | Per sample #Params$_{act}$ | FLOPs | Throughput (im/s) | IN-1K Accuracy |
|---|---|---|---|---|---|
| ConvNeXt-T (Liu et al., 2022) | 28.6 | 28.6 | 4.5G | 814 | 82.1 |
| ConvNeXt-T-4 Last 2 Top 1 | 34.5 | 25.6 | 4.2G | 768 | 82.1 |
| ConvNeXt-S (Liu et al., 2022) | 50 | 50 | 8.7G | 466 | 83.1 |
| ConvNeXt-S-4 Last 2 Top 1 | 56.1 | 47.3 | 8.5G | 442 | 83.1 |
| ConvNeXt-B (Liu et al., 2022) | 88.6 | 88.6 | 15.4G | 299 | **83.8** |
| ConvNeXt-B-4 Last 2 Top 1 | 99.1 | 83.4 | 15.0G | 289 | 83.5 |
| ConvNeXt-S *(iso.)* | 22.3 | 22.3 | 4.3G | 1100 | 79.7 |
| ConvNeXt-S-8 *(iso.)* Last 2 Top 1 | 38.9 | 22.3 | 4.3G | 1031 | **80.3** |
| ConvNeXt-B *(iso.)* | 82.4 | 82.4 | 16.9G | 336 | **82.0** |
| ConvNeXt-B-8 *(iso.)* Last 2 Top 1 | 115.4 | 82.4 | 16.9G | 303 | 81.6 |
| ViT-S | 22.0 | 22.0 | 4.6G | 1083 | 79.8 |
| ViT-S-8 Last 2 Top 2 | 38.6 | 25.0 | 5.3G | 892 | 80.5 |
| ViT-S-8 Every 2 Top 2 | 71.7 | 33.1 | 6.9G | 724 | **80.7** |
| ViT-B | 86.6 | 86.6 | 17.5G | 329 | **82.8** |
| ViT-B-8 Every 2 Top 2 | 284.9 | 129.9 | 26.3G | 227 | 82.5 |

Table 2: Accuracy for ImageNet-21K pre-trained models and different adaptations with ImageNet-1K: *IN-1K* is standard fine-tuning, *Linear prob* is the training of a linear layer for the output probability layer (everything else being frozen), and *0-shot* is the direct application of the pre-trained model to ImageNet-1K. "Every 2" and "Last 2" are the corresponding MoE strategies (see text). "Top $k$" corresponds to the number of experts involved.

| Architecture | #Params $(\times 10^6)$ | Per sample #Params$_{act}$ | FLOPs | IN-1K Accuracy | Linear prob Accuracy | 0-shot Accuracy |
|---|---|---|---|---|---|---|
| ConvNeXt-T (Liu et al., 2022) | 28.6 | 28.6 | 4.5G | 82.9 | 81.8 | 44.3 |
| ConvNeXt-T-8 Last 2 Top 1 | 70.0 | 28.7 | 4.5G | **83.5** | **82.3** | **44.6** |
| ConvNeXt-S (Liu et al., 2022) | 50.3 | 50.3 | 8.7G | 84.6 | 83.2 | 45.2 |
| ConvNeXt-S-8 Last 2 Top 1 | 91.6 | 50.3 | 8.7G | **84.9** | **83.6** | **45.3** |
| ConvNeXt-B (Liu et al., 2022) | 88.6 | 88.6 | 15.4G | **85.8** | **84.9** | 45.8 |
| ConvNeXt-B-8 Last 2 Top 1 | 162.0 | 88.6 | 15.4G | 85.7 | 84.8 | **45.9** |
| ViT-S | 22.0 | 22.0 | 4.6G | 82.6 | 81.7 | 44.2 |
| ViT-S-8 Every 2 Top 2 | 71.7 | 33.1 | 6.9G | **83.0** | **81.9** | **44.7** |
| ViT-B | 86.6 | 86.6 | 17.5G | 85.2 | 84.0 | **45.7** |
| ViT-B-8 Every 2 Top 2 | 284.9 | 129.9 | 26.3G | 85.2 | 84.0 | 45.6 |

## 4.2 Base results on ImageNet

Tab. 12 presents the results obtained on ImageNet-1k validation set by a model that has been entirely trained on ImageNet-1k, for isotropic architecture (e.g., ViT, ConvNeXt *iso.*) and a hierarchical architecture, namely ConvNeXt. We see some significant improvements in accuracy compared to the "no-MoE" results for small model sizes, especially for anisotropic models.

Tab. 2 presents the results of models that are pre-trained on ImageNet-21k, and tested on the same ImageNet-1k validation set than above. Here, MoE does bring some improvement for moderate numbers of activations per sample. However, this improvement decreases for large numbers of activations per sample, as depicted in Fig. 2.

> **MoE Strengthens Moderate Model Size; Not the Frontier**
>
> MoE integration provides clear gains at a moderate scale, with tiny and small models benefiting most. As data scale grows, these relative improvements strengthen. Yet as model capacity rises, these benefits diminish, showing that MoE enhances mid-sized models but does not redefine SoTA performance.

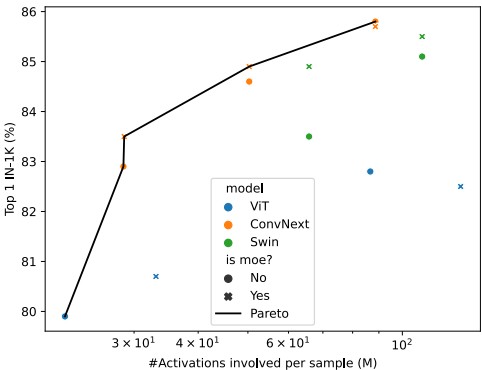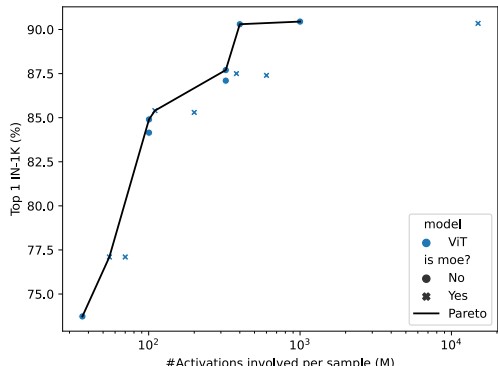

Figure 2: **Left**: Pareto-front for ImageNet21k, x-axis = number of activations per sample. ViT models have been presented in MoE versions only after additional pretraining, and are therefore not presented. MoE seems to be Pareto optimal for a number of activations per sample below 90M. **Right**: Pareto-front for ViT models on JFT-300M. Overall, MoE is never validated for a number of activations per sample above $\approx 100$M.

## 4.3 Sensitivity with Respect to Design Choices

This section presents different sensitivity studies that investigate the specifics of the design choices of the MoE layers. We explore the impact of the position of these layers inside the architecture (Sec. 4.3.1), the effect of varying the number of experts (Sec. 4.3.2), and finally, we assess the necessary design changes when transitioning from the ImageNet-1k to ImageNet-21k dataset (Sec. 4.3.4) and the impact of the routing architecture choices (Sec. 4.3.5). Ideally, sensitivity analyses would explore the joint effects of multiple factors (such as the number, placement, and size of experts), but exhaustive multi-axis exploration quickly becomes computationally intractable. Nevertheless, we partially address such interactions by analyzing the combined impact of expert size and expert count in **??**, highlighting how these factors jointly influence efficiency and accuracy.

### 4.3.1 Impact of the Position of MoE Layers

Table 3: Comparative results for different positions of MoE layers: ImageNet-1k training on ConvNeXt-T and ViT-S, all employing 8 experts.

| Architecture | MoE | # Params | | FLOPs | IN-1K |
| --- | --- | --- | --- | --- | --- |
| | | $\times 10^6$ | Sample | | |
| ConvNext-T | **no MoE** | 28.6 | 28.6 | 4.5G | **82.1** |
| ConvNext-T-8 | every 2 | 54.3 | 17.0 | 3.8G | 81.8 |
| | **stage** | 47.4 | 25.5 | 4.0G | **82.1** |
| | **last 3** | 49.9 | 25.0 | 4.1G | **82.1** |
| | **last 2** | 46.3 | 25.6 | 4.2G | **82.1** |
| ConvNeXt-S *(iso.)* | no moe | 22.3 | 22.3 | 8.7G | 79.7 |
| ConvNeXt-S-8 *(iso.)* | every 2 | 96.7 | 22.3 | 8.7G | 79.6 |
| | **last 2** | 38.9 | 22.3 | 8.7G | **80.3** |
| ViT-S | no MoE | 22.0 | 22.0 | 4.2G | 79.9 |
| ViT-S-8 | **every 2** | 71.7 | 33.1 | 5.3G | **80.7** |
| | last 2 | 38.6 | 25.0 | 6.9G | 80.5 |

¹

Tab. 3 displays the effects of various placements of MoE layers, comparing the following configurations: (a) **Every 2**: a MoE layer replaces every second layer; (b) **Stage**: a MoE layer replaces the final layer of each stage, resulting in four MoE layers throughout the network for ConvNeXt; (c) **Last 2**: a MoE layer replaces the final layer of each of the last two stages; (d) **Last 3**: Same as "Last 2", but with an additional MoE layer

Table 4: Comparative results for different numbers of experts. ImageNet-1k training using the "Last 2" strategy.

| Architecture | # experts | Top-k | #Params ($\times 10^6$) | Per samples #Params$_a$ | IN-1K Top 1 acc. |
|---|---|---|---|---|---|
| ConvNeXt-T | **no MoE** | - | 28.6 | 28.6 | **82.1** |
| ConvNeXt-T | **4** | 1 | 54.3 | 25.5 | **82.1** |
| | **8** | **1** | 47.4 | 25.5 | **82.1** |
| | 8 | 2 | 47.4 | 25.5 | 82.0 |
| | 16 | 1 | 46.3 | 25.5 | 81.7 |
| ViT-S | no MoE | - | 22.0 | 22.0 | 79.9 |
| ViT-S | 4 | 2 | 29.1 | 25.0 | 79.8 |
| | 8 | 1 | 38.6 | 22.0 | 80.2 |
| | **8** | **2** | 38.6 | 25.0 | **80.5** |
| | 16 | 2 | 57.5 | 25.0 | 80.2 |

Table 5: ImageNet-21k training on ConvNeXt-T employing 8 experts and an MLP ratio of 4 unless specified explicitly.

| MoE strategy | #Params ($\times 10^6$) | Per samples #Params$_{act}$ | FLOPs | IN-1K Top 1 acc. |
|---|---|---|---|---|
| no MoE | 28.6 | 28.6 | 4.5G | 82.9 |
| every 2 | 97.5 | 28.6 | 4.5G | 82.9 |
| **stage** | 78.2 | 28.6 | 4.5G | **83.5** |
| **last 3** | 70.0 | 28.6 | 4.5G | **83.5** |
| **last 2** | 70.0 | 28.6 | 4.5G | **83.5** |
| last 2 mlp_ratio 2 | 46.3 | 25.6 | 4.2G | 83.4 |

inserted in the middle of stage 3. As shown in Tab. 3, "Last 2" strategy is the most robust choice: it performs well across all architectures. In contrast, "Every 2" performs worst for ConvNeXt architecture, and the best for ViT. Consequently, in the following, all experiments use "Last 2" for ConvNeXt and "Every 2" for ViT.

> **MoE placement matters**
>
> While ViT benefits most from inserting MoE layers at every second block and ConvNeXt from placing them in the last two stages, the Last 2 strategy offers a simple, broadly effective rule of thumb that consistently delivers strong performance across architectures.

### 4.3.2 Influence of the Number of Experts

Tab. 4 presents the performance of ConvNext and ViT architectures on ImageNet-1K training with the "Last 2" strategy MoE layers. It provides a comprehensive view of how the number of experts influences the size (parameter count) of the models and their Top-1 accuracy on this dataset.

The experimental results suggest that four experts yield the best performance for ConvNeXt, while eight experts are optimal for ViT. This is showcased by the Top-1 accuracy rates, which are consistent at 82.1% for ConvNeXt with four experts and reach a peak of 80.5% for ViT with eight experts. However, we note that increasing the number of experts to 16 has a detrimental effect on both architectures. Specifically, for ConvNeXt, the top-1 accuracy slightly drops to 81.7%, and for ViT, the performance plateaus at 80.2%.

These findings highlight that while MoE layers can enhance model performance, there is a delicate trade-off to be struck in terms of the number of experts. Exceeding the optimal number can lead to suboptimal results, negating the potential benefits of the MoE integration. Furthermore, in our detailed analysis, described in Sec. 4.3.3, we explore the interplay between the number and the size of each expert. Our investigation revealed that, for isotropic networks such as ViT and ConvNeXt *iso.*, reducing the size of experts adversely affects performance, while for ConvNeXt, it has no significant impact. Moreover, during the course of our experiments, we observed that while the Top-1 configuration was superior for ConvNeXt, the Top-2 configuration for ViT was comparable or even better than the ConvNeXt Top-1 configuration, leading us to employ Top-1 for ConvNeXt and Top-2 for ViT (cf. Tab. 4).

> **Expert Count under Moderate Data Scale**
>
> On ImageNet-1k, ConvNeXt models reach peak performance with 4 experts, while ViT benefits from up to 8. This contrast stems from architectural width: ConvNeXt's broad final stages saturate quickly, limiting the value of additional experts, whereas ViT's narrower last stage tolerates more experts.

### 4.3.3 Impact of Expert Size and Number

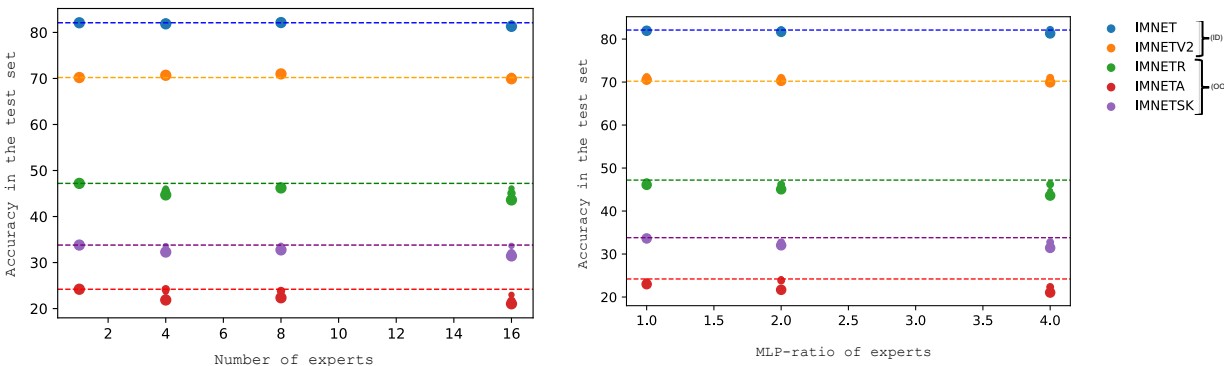

(a) # experts ($\in \{1, 4, 8, 16\}$) vs. performance. **Points' size**: MLP-ratio ($\in \{1, 2, 4\}$).

(b) Experts' MLP-ratio ($\in \{2, 4\}$) vs. performance. **Points' size**: # experts ($\in \{1, 4, 8, 16\}$).

Figure 3: Exploring the interplay between the size and count of experts in a MoE layer for ConvNeXt-T on ImageNet-1K. Baseline results (without MoE) are denoted by dotted lines. For this small dataset, MoE does not bring much improvement (see Fig. 2 for bigger datasets).

Two critical tunable parameters in the MoE layers are the number of experts per layer and the size of each expert. The latter is defined by the mlp_ratio, which scales the hidden dimension of the feed-forward network (FFN) (Sec. 3.1). Fig. 3 illustrates the sensitivity of ConvNeXt's performance to variations in these parameters, with the number of experts ranging from 4 to 16 and the mlp_ratio from 1 to 4. By default (original network), this ratio is set to 4, meaning that values below 4 correspond to smaller per-sample FFN capacities than in the original network.

To assess generalization, we report top-1 accuracy across three dataset categories: (i) in-distribution (ID) datasets—ImageNet and ImageNetV2 (Recht et al., 2019); (ii) out-of-distribution (OOD) datasets—ImageNet-R (Hendrycks et al., 2021a) and ImageNet-Sketch (Wang et al., 2019); and (iii) adversarial datasets—ImageNet-A (Hendrycks et al., 2021b). The objective is to determine whether performance benefits more from using many smaller experts or fewer larger ones.

Our experiments on ConvNeXt reveal that while MoE layers can slightly improve or preserve performance on ID datasets, they tend to degrade performance on OOD datasets. In particular, increasing the number of experts to 16 consistently reduces accuracy across all dataset types. Regarding expert size, smaller experts appear marginally more robust to increases in expert count. Overall, the best tradeoff for ID is achieved with 4 to 8 experts and an mlp_ratio of 2 or 4. Based on these findings, we adopt a configuration of 4 experts with an mlp_ratio of 2 for experiments on ImageNet-1K. For ConvNeXt isotropic and ViT architectures, however, reducing the mlp_ratio consistently harms performance, and in the case of ViT, it also introduces training instabilities. Also, OOD performance is typically degraded in Fig. 3.

> **Balancing Expert Number and Width on ImageNet-1k**
>
> On ImageNet-1k, the best trade-off is achieved with 4–8 experts and mlp_ratio 2–4; increasing to 16 experts consistently reduces accuracy and OOD robustness. Increasing number of experts while reducing their size helps reducing overfitting.

### 4.3.4 Results on ImageNet-21k

Tabs. 5 and 6 present the results of models trained on ImageNet-21k, reporting ImageNet-1k accuracy after fine-tuning. We analyse the effect of MoE layer positioning, expert size (Tab. 5), and expert count (Tab. 6). The results demonstrate that, when large volumes of data are accessible, a greater number of experts can be effectively deployed. Notably, the use of sixteen experts, which was sub-optimal for ImageNet-1k, does

Table 6: ImageNet-21k training on ConvNeXt-T: comparison between various numbers of experts, with an MLP ratio of 4.

| # experts | #Params $(\times 10^6)$ | Per samples #Params$_{act}$ | FLOPs | IN-1K Top 1 acc. |
|---|---|---|---|---|
| no MoE | 28.6 | 28.6 | 4.5G | 82.9 |
| 8 | 54.3 | 28.6 | 4.5G | 83.5 |
| **16** | 117.1 | 28.6 | 4.5G | **83.6** |
| 32 | 211.6 | 28.6 | 4.5G | 83.4 |

not negatively affect performance on ImageNet-21k. This suggests that increasing the number of experts, together with the volume of data, could lead to valuable enhancements.

In terms of MoE layer configuration, our data showed no performance improvements with the addition of more layers, and the "every 2" strategy again yielded the poorest results for ConvNeXt-based architecture.

> **Scaling Expert Count with Larger Pretraining**
>
> On ImageNet-21k, ConvNeXt scales up to 16 expert,s and ViT also benefits from more. Increased data reduces overfitting, allowing more experts. This highlights the link between dataset scale and MoE capacity. The optimal layer positioning remains unchanged.

### 4.3.5 Impact of the Routing Architectures

We investigated three distinct routing architectures for our study:

- "Conv": A straightforward $1 \times 1$ convolution, similar to a linear gate.

- "Cos": A gate that utilizes cosine similarity for routing, as detailed in (Chi et al., 2022). It is a two-layer architecture, with one linear layer projecting the features to a lower-dimensional space and another performing cosine similarity against some learned latent codes.

- "L2": This is identical to the "Cos" gate, except that it employs the L2-distance for similarity instead of cosine.

We conducted training on the ImageNet-1K dataset using various small networks, each employing different routing mechanisms as displayed earlier. The results, as shown in Tab. 7, indicate that the simple convolutional (conv) configuration generally yields slightly better performance in most cases. However, when evaluating these models on out-of-distribution (OOD) datasets, the results are more varied. The inclusion of OOD evaluation is crucial, as an effective routing mechanism should ideally facilitate an expert decomposition that generalizes well. Notably, the MoE models that showed superior performance on ImageNet-1K also displayed enhanced robustness in the OOD evaluations.

Table 7: Performance of MoE-Integrated Models vs Non-MoE Baselines: ablation study on the gating network architecture. MoE layers are placed on the last two even layers.

| ConvNeXt-S | | | | |
|---|---|---|---|---|
| Gate | IN-1K | IMV2 | IMA | IMR | IMSK |
| no MoE | **82.1** | 70.8 | 24.2 | **47.2** | **33.8** |
| Conv | **82.1** | **71.0** | 23.8 | 46.2 | 32.7 |
| Cos | 81.9 | 70.6 | 22.3 | 45.6 | 33.0 |
| L2 | 82.0 | 71.6 | **25.0** | 45.9 | 32.8 |
| **ConvNeXt-S *(iso.)*** | | | | |
| Gate | IN-1K | IMV2 | IMA | IMR | IMSK |
| no MoE | 79.7 | 68.6 | 13.0 | 46.4 | 34.0 |
| Conv | **80.3** | **68.8** | 13.5 | **46.6** | **34.4** |
| Cos | 79.7 | 68.4 | 12.2 | 45.3 | 33.0 |
| L2 | 80.1 | 68.6 | 13.3 | 46.1 | 34.3 |
| **ViT-S** | | | | |
| Gate | IN-1K | IMV2 | IMA | IMR | IMSK |
| no MoE | 79.8 | **69.1** | 19.8 | **43.4** | 29.7 |
| Conv | **80.5** | **69.1** | **20.3** | 43.2 | **29.9** |
| Cos | 79.7 | 67.8 | 15.9 | 41.6 | 29.3 |
| L2 | 80.2 | 68.8 | 19.0 | 41.9 | 29.2 |

evaluation is crucial, as an effective routing mechanism should ideally facilitate an expert decomposition that generalizes well. Notably, the MoE models that showed superior performance on ImageNet-1K also displayed enhanced robustness in the OOD evaluations.

> **Simplicity Wins in Routing Architectures**
>
> Across both ConvNeXt and ViT, a straightforward linear gate achieves the best overall results. More complex routing adds no clear benefit and often underperforms, even on OOD data.

Table 8: Robustness of MoE w.r.t. to domain change, for two different pretrainings, both fine-tuned on ImageNet-1k: Top-1 accuracies on various datasets.

| Model | IN-1K | IMV2 | IMA | IMR | IMSK |
|---|---|---|---|---|---|
| **Imagenet1k trained** | | | | | |
| ConvNeXt-T | **82.1** | 70.8 | **24.2** | **47.2** | **33.8** |
| ConvNeXt-T-4 | **82.1** | **71.0** | 23.8 | 46.2 | 32.7 |
| ConvNeXt-S | **83.1** | **72.5** | **31.3** | **49.6** | **37.0** |
| ConvNeXt-S-4 | **83.1** | 72.2 | 30.2 | 49.0 | 37.3 |
| ConvNeXt-B | **83.8** | **73.4** | **36.7** | **51.3** | **38.2** |
| ConvNeXt-B-4 | 83.5 | 72.8 | 33.9 | 48.6 | 36.6 |
| ViT-S | 79.9 | 68.8 | 19.8 | 43.4 | 29.7 |
| ViT-S-8 | **80.7** | **70.1** | **21.1** | **43.9** | **30.9** |
| ViT-B | **82.8** | **72.1** | **32.4** | **51.2** | **36.9** |
| ViT-B-8 | 82.5 | 71.5 | 32.0 | 46.6 | 35.2 |
| ConvNeXt-S *(iso.)* | 79.7 | 68.6 | 13.0 | 46.4 | 34.0 |
| ConvNeXt-S-8 *(iso.)* | **80.3** | **68.8** | **13.5** | **46.6** | **34.4** |
| ConvNeXt-B *(iso.)* | **82.0** | **71.1** | **21.2** | **50.0** | **38.1** |
| ConvNeXt-B-8 *(iso.)* | 81.6 | 70.6 | 19.1 | 48.5 | 35.7 |
| **Imagenet21k pretrained** | | | | | |
| ConvNeXt-T | 82.9 | 72.4 | **36.2** | 51.1 | 38.5 |
| ConvNeXt-T-8 | **83.5** | **72.8** | 32.6 | **51.3** | **40.8** |
| ConvNeXt-S | 84.6 | 74.7 | **44.8** | 57.5 | 43.6 |
| ConvNeXt-S-8 | **84.9** | **75.5** | 44.4 | 55.7 | **45.5** |
| ConvNeXt-B | **85.8** | 76.0 | **54.6** | **62.0** | **48.8** |
| ConvNeXt-B-8 | 85.7 | **76.3** | 51.8 | 59.6 | 48.4 |
| ViT-S | 82.6 | 72.6 | **38.9** | 50.8 | 39.0 |
| ViT-S-8 | **83.0** | **72.7** | 35.3 | **51.0** | **39.4** |
| ViT-B | 85.2 | **76.1** | **56.0** | **61.5** | **46.9** |
| ViT-B-8 | 85.2 | 75.4 | 48.1 | 59.3 | 45.9 |

# 5 Discussion

## 5.1 Hierarchical vs Isotropic Models

First, one can note that the best MoE designs typically apply experts only to the last layers. ViT operates at the same resolution at each layer after the input images have been cut into patches. This isotropic design implies that MoE is applied in layers that are not particularly large. On the other hand, with hierarchical models such as ConvNeXt, applying MoE to the last layers considerably increases model size. Indeed, as presented in Tab. 6, employing eight experts doubles the size of the network despite using only two layers of MoE.

## 5.2 Positions and Numbers of MoE Layers

There are two prevalent strategies in the literature concerning the position of MoE layer: "Every 2" and "Last 2". In the context of hierarchical architectures such as ConvNeXt, the "Every 2" strategy dramatically increases the number of weights while simultaneously yielding inferior results. However, for ViT "Last 2" and "Every 2" are two viable strategies. This is in line with the results of V-MoE (Riquelme et al., 2021) on large-scale datasets. Overall, our results confirm that "Last 2" is a solid starting point, as it consistently yields positive outcomes across all tested architectures. However, it may not always be the optimal choice for every architecture. For instance, in the case of ViT, "Every 2" demonstrates better results. Regarding the number of experts, we noted that model performance quickly saturates with the number of experts, as reproduced in Tab. 6, and detailed in Fig. 3.

## 5.3 Robustness Evaluation

We evaluated MoE models, trained or fine-tuned on ImageNet-1k, against a range of robustness benchmark datasets, such as ImageNet-A (IMA) (Hendrycks et al., 2021b), ImageNet-R (IMR) (Hendrycks et al., 2021a),

Table 9: ViT-B trained on ImageNet-1k with different hyperparameters. In particular, for the top row (gray) we follow the training recipe from Deit (Touvron et al., 2021) and for the bottom one we follow Deit III (Touvron et al., 2022).

| Architecture | # Params ($\times 10^6$) | Per sample #Params$_{act}$ | IN-1K Acc. |
|---|---|---|---|
| ViT-B-8 (Touvron et al., 2021) | 86.6 | 86.6 | 81.8 |
| ViT-B-8 Every 2 Top 2 | 284.9 | 129.9 | **82.2** |
| ViT-B (Touvron et al., 2022) | 86.6 | 86.6 | **82.8** |
| ViT-B-8 Every 2 Top 2 | 284.9 | 129.9 | 82.5 |

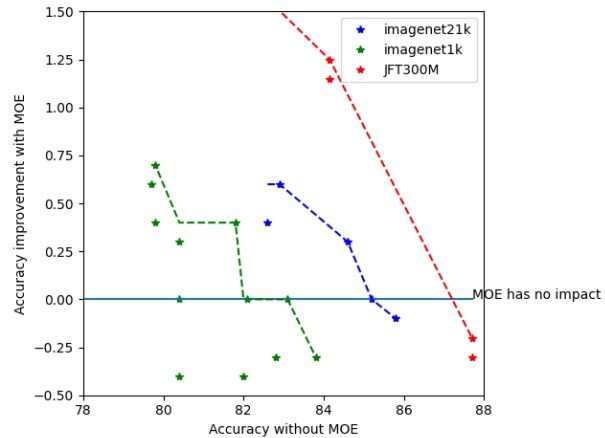

Figure 4: Improvement from MoE (y-axis) vs. baseline accuracy without MoE (x-axis) for image classification, across pretraining sizes. Larger pretraining yields greater gains; higher baseline accuracy reduces impact. Dashed lines show maximal improvement per accuracy level. Results are drawn from Tabs. 2, 11 and 12 (in App. B).

ImageNet-Sketch (IMSK) (Wang et al., 2019), and ImageNet V2 (IMV2) (Recht et al., 2019), the latter being used as a measure of overfitting.

Tab. 8 reports Top-1 accuracy for all datasets. It shows that MoE models tend to underperform their dense counterparts in ImageNet-1k. However, we note that MoE models are better for the robustness metric; they were actually already better on ImageNet-1k. So, these results actually emphasize the success of MoE when the model is sufficiently small to be improved by a MoE, as discussed in the previous section: the domain generalization holds only in the regime in which MoE improves the base results.

### 5.4 On Data-Augmentation

As shown in the previous section, MoE is sensitive to the amount of data at hand. This sensitivity stems from the inherent structure of MoE, where increasing the number of experts results in each expert operating on a smaller data fraction. Consequently, each expert tends to learn specific data clusters, as discussed by (Chen et al., 2022).

Modern training methodologies often employ robust data augmentation techniques to enhance model generalization. However, the interplay between these augmentations and the data clusters learned by the MoE model remains relatively uncharted. For instance, techniques like Mixup (Zhang et al., 2018), which blend images, could blur the distinctions between these clusters. Similarly, the Random Resize method, with its aggressive magnification capability, may distort the original data clusters. Understanding the precise influence of these augmentations on the MoE model's learning process is essential, as it holds implications for the model's ultimate performance. Some evidence of this can be seen in Tab. 9. While one training recipe might show MoE outperforming its dense counterparts (as indicated in the top rows of the table), a different, more effective recipe could lead to the opposite outcome (Bottom rows of the table).

### 5.5 Alternate Views

With the ImageNet benchmark increasingly reaching saturation, the focus in contemporary computer vision models has shifted towards computational efficiency and scaling prowess. An essential question arises: *Does the MoE model enhance this aspect in image classification?* Our experiments with ImageNet reveal an intriguing insight: integrating MoE into models like ConvNeXt and ViT enhances performance, but this

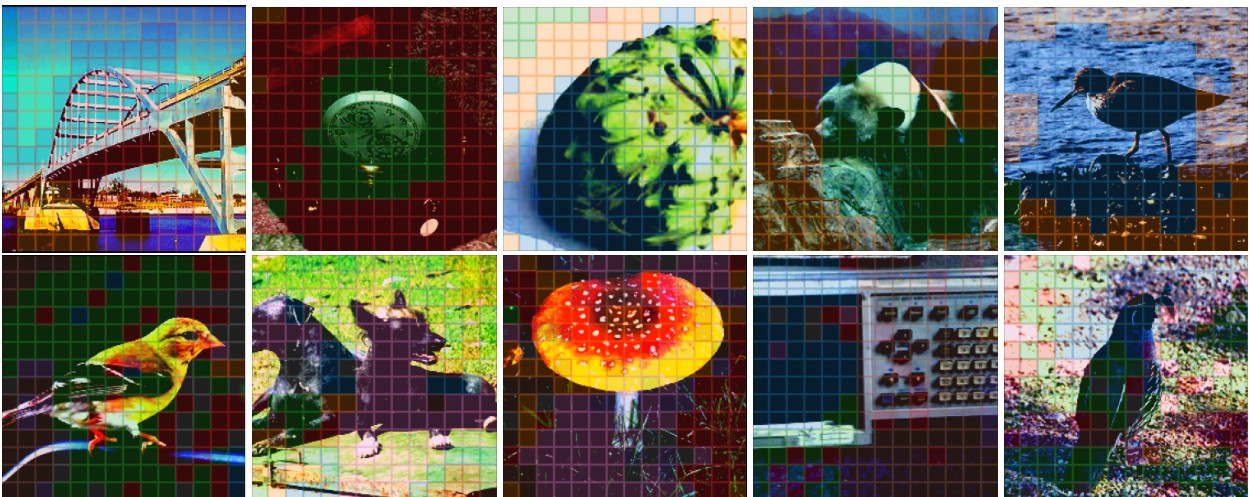

Figure 6: The top row features the ConvNext-T-4 model, while the bottom row showcases the ViT-S-8 model, both of which have been trained on the ImageNet1k dataset. The images displayed have been generated by upsampling the gating output, retaining only the top expert for clarity. Each distinct color represents a unique expert. These specific images are samples from the ImageNet validation set, extracted from the last third stage of the ConvNeXt-T-4 and the final layer of the ViT-S-8 models, respectively.

improvement tends to plateau in models with over 100M parameters. At first glance, this might seem to contradict findings from some papers (Riquelme et al., 2021; Mustafa et al., 2022). However, with additional context, this observation aligns with the existing literature, as evidenced by results in Figs. 2 and 4, and Tab. 11 (App. B). Specifically, as the base dense network grows larger, the performance gap between MoE and its dense counterpart narrows.

Exploring this further in large-scale datasets, such as JFT 3 billion (Zhai et al., 2022), reveals similar findings. For instance, the vision transformer model SoViT, with 400M parameters (Alabdulmohsin et al., 2023), exhibits comparable performance to the 15B V-MoE (Riquelme et al., 2021) when both use ViT-based architectures. Similarly, when considering the JFT-3B dataset (Zhai et al., 2022), the 5.6B LiMoE-H model (Mustafa et al., 2022) and Coca-Large (Yu et al., 2022) lead to analogous results when evaluated on a per-sample parameter basis. Interestingly, in the vision-language modeling domain, Lin et al. (2024) observe that increasing the number of experts for the image modality does not lead to diminishing returns, aligning with our findings regarding the efficacy of experts in scaling performance.

To summarize, while the MoE model does not appear to redefine the state-of-the-art when matched against dense models on both ImageNet and billion-scale datasets, its real strength emerges in enhancing smaller models. By integrating MoE, these compact models can achieve better performance, pushing their capabilities closer to the forefront of current benchmarks. Yet, it is essential to acknowledge that, in the broader context, MoE might not drive significant advancements in the overall state-of-the-art. Moreover, shape optimization as demonstrated in SoViT, could further improve performance and could even compound with the benefits of MoE, enabling smaller and more capable models. This nuanced perspective is further enforced in the appendix, with the data showcased in Tab. 11 in App. B.

### 5.6 Model Inspection

The objective of this section is to examine the routing component of the model. We will present the top expert for each routing layer. Note that when considering the top-2 configuration, we only report, for this visualization, the top-1 of these two experts.

**Visual inspection:** Fig. 6 provides a visualization of the portions of images that are assigned to the different experts. This reveals that some

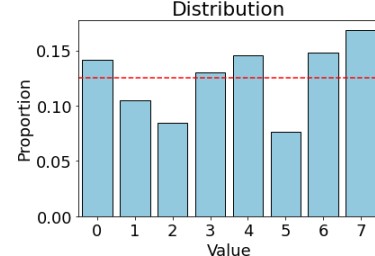

Figure 5: Distribution of experts

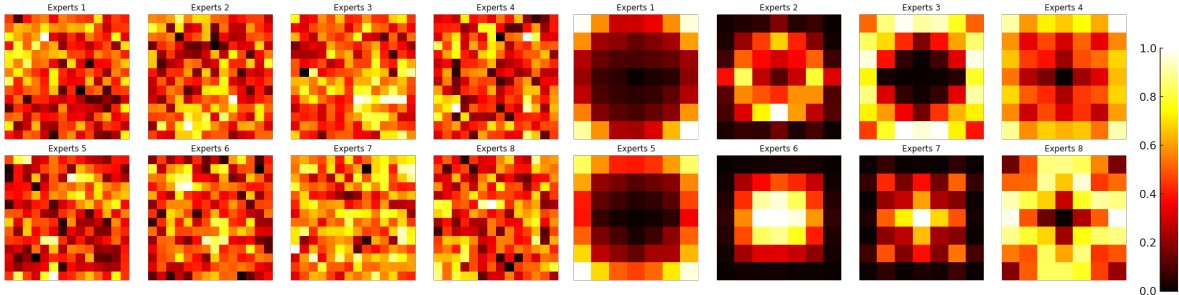

(a) ViT-B deepest MoE. No absolute positional bias is observed.

(b) ConvNeXt-B last MoE layer. A clear spatial partitioning of experts emerges.

Figure 7: Average spatial distribution of experts trained on ImageNet-22K.

experts specialize in specific elements of an image, such as animals, buildings, or objects. This trend is especially apparent when considering the top-1 MoE, but the distinction becomes less clear with the top-2 MoE. In both instances, the clusters formed by each expert are challenging to interpret, and the spatial locations of each expert tend to lack clarity and can be quite ambiguous. However, when directly examining the spatial distribution, a clear spatial decomposition emerges for ConvNeXt models (cf. Fig. 7). In particular, Fig. 7b shows that certain experts focus specifically on the image borders or the center. This behavior may be explained by the strong local spatial bias imposed by convolution, in contrast to the attention-based models, which do not exhibit such patterns.

**Number of experts involved per image:** We quantified the average number of experts contributing to processing an image. In the initial layers, the majority of experts are typically engaged for each image. With fewer than eight experts, most are utilized, even in deeper layers. For instance, in the ViT model, with eight experts, an average of seven experts contributed per image, while in the ConvNeXt model, there are six. As we increase the number of available experts per MoE layer, the number of contributing experts per image also rises. For example, in ConvNeXt, with 16 experts, an average of 10.5 experts are involved, and with 32 experts, this number increases to 15.8. This suggests that each expert is likely focused on a limited number of image patches. This observation is confirmed by the cumulative distribution function (CDF) of each expert per image, as depicted in Fig. 8. On the other hand, when directly examining the routing distribution across experts, we observe a well-balanced assignment, indicating that the load-balancing loss is effectively fulfilling its role, as depicted in Fig. 5.

**Correlation between experts and labels:** Fig. 9 (top) presents the correlation between the number of expert occurrences and individual classes in ImageNet-1k. The class IDs in ImageNet-1k are organized in such a way that adjacent classes often share similar attributes. For instance, class IDs ranging from 151 to 268 are all dedicated to different dog breeds. From Fig. 9, we note that the occurrences in the first three MoE layers do not match the ImageNet-1k classes, with most experts appearing uniformly across different classes. However, in the last three MoE layers, there is a noticeable trend of more experts aligning with specific ImageNet-1k classes. This distinction is especially pronounced between experts focusing on animals (classes up to ID 397, such as experts 0 and 6 in Fig. 9-(e)) and those centered on objects (classes beyond ID 397, like experts 1 and 5 in Fig. 9-(e)). As the number of experts increases, this class-specific alignment still holds, though it remains challenging to clearly define the roles of these experts since they often span several scattered classes rather than forming tight, contiguous groups that would fit each class, as illustrated in Fig. 10.

**Expert Similarity:** In order to quantify the similarity between experts, we use a method derived from the Jaccard index. For each image, we count the occurrences $c_i$ of each expert $E_i$. The similarity between two experts, $E_i$ and $E_j$, is defined as $S_{ij} = \frac{|E_i \cap E_j|}{|E_i \cup E_j|} = \frac{\sum^{E_i \cap E_j} c_i + c_j - |c_i - c_j|}{\sum^{E_i \cup E_j} c_i + c_j}$, where $E_i \cap E_j$ represents the images where both $E_i$ and $E_j$ appear together, while $E_i \cup E_j$ refers to all images where either $E_i$ or $E_j$ is present at least once. The absolute value term $|c_i - c_j|$ acts as a corrective factor to account for scenarios where one expert may be deployed for a minimal number of patches while the other is utilized for the majority,

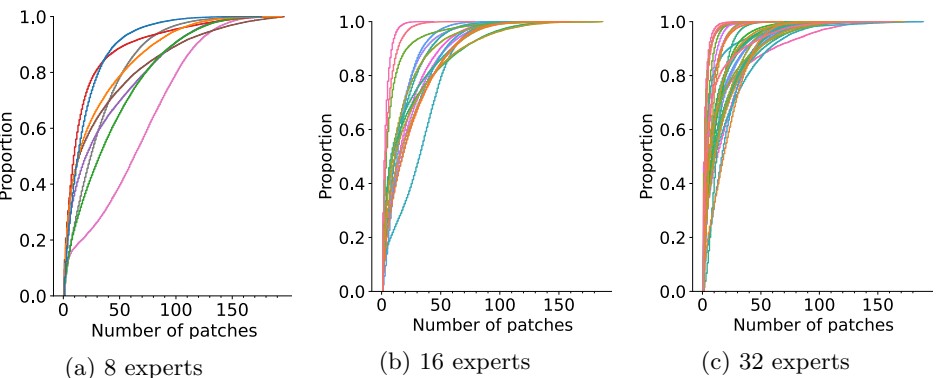

(a) 8 experts        (b) 16 experts        (c) 32 experts

Figure 8: Cumulated Distribution Function of the number of patches used per image for ConvNeXt-S trained on ImageNet-21k, with 8, 16, and 32 experts. Each colour corresponds to one expert. Sample detailed interpretation: for 8 experts, the "red" expert sees between 1 and 30 patches (out of 196) for 80% of the images on which it is active. In particular (and even more so when there are many experts), an expert frequently covers a limited part of the image. For instance, if the y-axis is close to 1 at x-axis=100 patches, covering more than 100 patches over 196 is rare: experts don't specialize much. As the number of experts increases, this tendency of experts being used only on a few patches increases too. For 32 experts, most of them focus on less than 30 patches; 80% of the time, they are active.

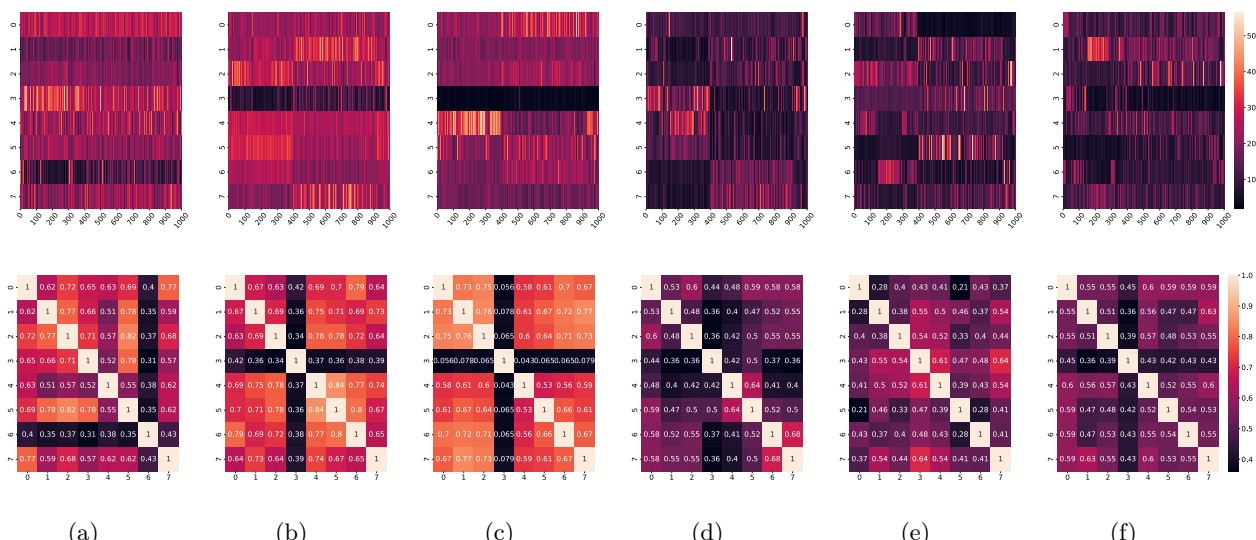

(a)      (b)      (c)      (d)      (e)      (f)

Figure 9: Routing analysis of ViT-S-8, trained on ImageNet-1k. Top row: per-class occurrence of the experts, with the abscissa indicating the class ID and the ordinate representing the expert index (8 experts). Bottom row: similarity score between the experts. The graphs are arranged from left to right from the MoE layer closest to the input (a) to the layer closest to the output (f).

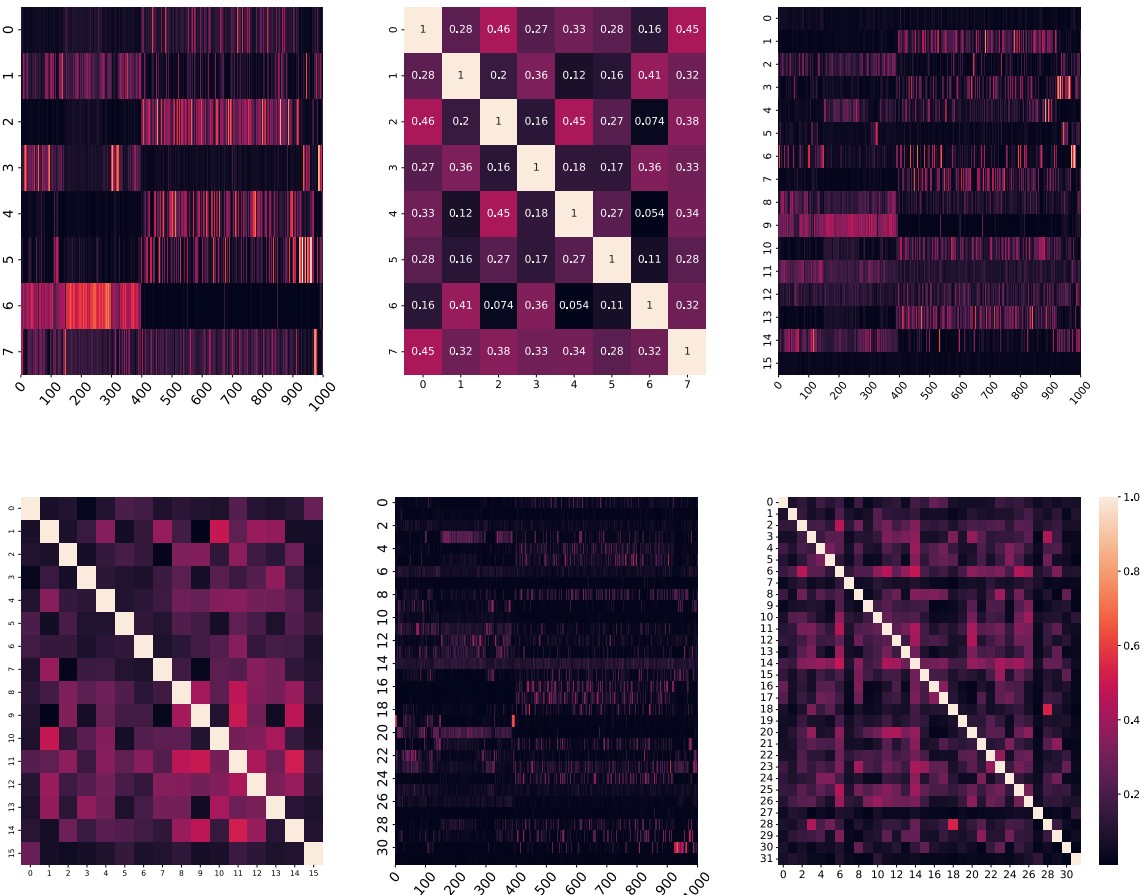

Figure 10: ConvNext-S trained on ImageNet-21k with 8, 16, and 32 experts trained on ImageNet-21k routing analysis. The right columns display the frequency of each expert across different classes, where the x-axis represents class IDs, and the y-axis denotes the expert number. The left column provides a similarity score between experts.

indicating a lack of synergy between the two experts. In the bottom panel of Fig. 9, we observe a distinct pattern in the utilization of experts across the stages of the MoE. In the initial layers, there is a prevalent co-occurrence of multiple experts, which highlights a concentration on local image features across all the images. Conversely, as we analyze the network deeper, the frequency of expert co-occurrence diminishes. This shift indicates a transition in focus from local image elements to more semantically rich aspects, with individual experts concentrating on specific semantic attributes unique to certain images.

**Are experts well partitioned and clustered?** For a general task such as ImageNet-1k, there is no clear, natural decomposition into experts, making it difficult to precisely assess if one expert decomposition is better than another, aside from comparing final accuracy. This comparison, however, provides limited insight into the quality of the experts. The literature (Chen et al., 2023; Mittal et al., 2022; Mohammed et al., 2022) suggests that, without explicit enforcement, MoE models struggle to separate tasks or find natural clusters beneath them. For instance, when a task separation was explicitly incorporated into the loss, there was a marked improvement in both accuracy and the ease with which experts could be pruned for specific tasks (Chen et al., 2023). Similarly, Mittal et al. (2022) found that the enforced decomposition resulted in substantial gains, while the absence of such enforcement left the model unable to discover optimal partitions. Consequently, we make three observations on ImageNet. First, as the number of experts increases, there is

a corresponding rise in the number of experts actively involved. Even with 32 experts, almost half of the experts are used per image in the last MoE layers. This seems counterintuitive since ImageNet is centered around objects; at least for deep layers, it should plateau at some point. Instead, experts are not well aligned with classes and spread widely, as depicted in Fig. 9. Second, an analysis of the cumulative distribution function (CDF) of experts' engagement (Fig. 8) reveals a prevalent trend in which most experts are assigned to less than 20 patches of an image, equating to less than 10% of the total image area, 50% of the time. Moreover, these patches are not necessarily contiguous. Third, when we look at the visualization of each expert, like in Fig. 6, we observe a concentration of experts in the image for MoE layers near the output. The interpretation of each expert's role becomes challenging: An expert does not seem to adhere closely to an object, or to a part of the location of an object.

---

**Lack of Coherent Expert Partitioning**

Qualitative and quantitative analyses reveal no consistent expert partitioning in vision MoE, particularly as the number of experts increases. With few experts, routing weakly aligns with ImageNet classes, but as the scale grows, expert allocation becomes diffuse, many experts activate per image, each covering only a small portion of patches, and long-range consistency across class IDs becomes less clear. ConvNeXt shows some spatial biases, whereas ViT shows none; in both, assignments remain well load-balanced yet not semantically disentangled, suggesting that the observed gains stem from added capacity rather than interpretable specialization.

---

## 6   Conclusion

Implementing effective Mixture-of-Experts (MoE) models for image classification tasks remains a challenging and open problem. Our experiments show that, particularly in large-scale models, MoE for image classification offers limited gains over state-of-the-art methods in accuracy, robustness, generalization, or per-sample efficiency across both convolutional networks and vision transformers. This contrasts with NLP tasks, where MoE improves performance by enhancing capacity for knowledge storage and retrieval. In image classification, which relies more on feature extraction and extrapolation, MoE models show only modest benefits, mainly in smaller models with fewer activated parameters. These findings suggest that the effectiveness of MoE in image classification tasks is highly context- and scale-dependent, with advantages diminishing as model capacity grows.

### Acknowledgements

The authors thank the reviewers for their insightful and constructive feedback, which has substantially improved the quality of this work. AL acknowledges support from the French National Research Agency (ANR) under Grant No. ANR-23-CPJ1-0099-01.

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
