# OpenReview forum: "Mixture of Experts for Image Classification: What's the Sweet Spot?"
_TMLR — Accepted by TMLR_

### Review · Reviewer_t2so · 2025-07-04

**Summary Of Contributions:**

This paper presents a systematic empirical study on integrating Mixture-of-Experts (MoE) into image classification architectures, namely ConvNext and Vision Transformer (ViT), using the ImageNet-1K and ImageNet-21K datasets. The authors examine various design choices—including the number of experts, placement of MoE layers, and routing mechanisms—and report the performance-efficiency trade-offs associated with these decisions. The work aims to offer practical guidance for applying MoE to classification tasks with open datasets.

**Audience:**

Yes

**Claims And Evidence:**

No

**Requested Changes:**

The following changes are suggested to improve the clarity and rigor of the paper:

1. Improve organization and clarity of writing. As an empirical study, the paper would greatly benefit from a clearer structure such as:

- State the motivating questions;

- Summarize key findings upfront;

- Explain experimental setup concisely;

- Present results in focused sections;

- Discuss insights and practical implications.

2. Justify design decisions more thoroughly. For example, explain why “every 2,” “last 2,” and “last 3” configurations were selected for MoE placement. If a full sweep is impractical, at least justify the sampling.

3. Consider interactions between factors. Rather than isolating one design choice while keeping others fixed, analyze how combinations of choices (e.g., number of experts vs. MoE placement) affect performance and trade-offs.

4. Refine insights and reduce redundancy. Remove observations that are not actionable or insightful. Instead, focus on highlighting trends or guidelines that could inform future MoE designs.

5. Revisit the model inspection section. The visualizations are interesting but currently inconclusive. Consider integrating them with quantitative measures or user studies to make stronger claims about expert behavior.

A good and very relevant reference is [1]. The authors can learn from [1], and also need to justify the difference from [1] with larger contribution.

[1] Han X, Wei L, Dou Z, et al. ViMoE: An Empirical Study of Designing Vision Mixture-of-Experts[J]. arXiv preprint arXiv:2410.15732, 2024.

**Strengths And Weaknesses:**

### Strengths:

- The paper addresses a timely and practically relevant topic: manually optimizing MoE for image classification without requiring billion-scale datasets.

- It conducts a wide-ranging empirical analysis, covering multiple axes of design choices (e.g., layer position, number of experts, routing methods).

### Weaknesses:

- The paper suffers from significant structural and writing issues. The experimental findings are not clearly organized or summarized, making it difficult for readers to extract the key insights efficiently.

- The experimental design lacks rigor in selecting candidates for evaluated design aspect. For example, when studying layer positions, the selected configurations (e.g., “every 2,” “last 2,” “last 3”) appear arbitrary and biased, with no comprehensive enumeration or justification.

- There is limited analysis of the interaction between different design choices. For instance, the effect of expert count is analyzed independently from layer position, though these factors are likely interdependent.

- While the paper reports many observations, few are distilled into actionable or generalizable insights. As a result, the overall contribution is limited.

- The MoE inspection and visualizations, while informative, lack clear conclusions or takeaways, especially regarding the interpretability and functionality of the experts.

---

> ### Author Response · Authors · 2025-09-08
>
> We thank the reviewer for the thorough and constructive feedback. Below, we address the concerns raised and outline planned revisions.
>
> > “The paper suffers from significant structural and writing issues. ...”
>
> We appreciate this feedback. In the revised version, we will restructure the paper for clarity:
>
> (a) State the motivating questions at the start (e.g., Where should MoEs be placed? How many experts are optimal? Which routing schemes are robust?); (b) summarize the key findings upfront; (c) make experimental setup clearer in Section 4.1; and (d) present results along distinct axes (placement, expert count, routing), each ending with a clear takeaway message. This restructuring will make the empirical contributions more accessible to readers.
>
> > “The experimental design lacks rigor in selecting candidates for evaluated design aspect ...”
>
> We acknowledge that our placement configurations may appear ad hoc. In practice, they were guided by both prior work and pilot experiments:
>
> * “Every 2” mirrors the V-MoE design for ViTs, where dense and sparse layers alternate. This configuration indeed yielded the strongest results for ViTs in our experiments (see Table X), but not for ConvNeXt.
> * “Last 2 / Last 3 / Last stage” placements were motivated by the intuition that deeper stages encode more abstract features. With a simple linear gating function, these later representations are more likely to support meaningful expert specialization.
>
> These configurations were therefore chosen as the most practical candidates, balancing coverage with feasibility. Table 3 highlights that most of the performance improvement comes from adding MoE layers to the last two blocks, supporting the intuition that these layers are the most critical for MoE placement. We agree that a full sweep of all possible placements would provide a more exhaustive picture. Unfortunately, given the compute demands, we had to restrict our study to a tractable set of configurations. We will make this rationale explicit in the revision.
>
> > “There is limited analysis of the interaction between different design choices. For instance, the effect of expert count is analyzed independently from layer position, though these factors are likely interdependent.”
>
> We agree that interactions between design choices are important. Due to compute constraints, we primarily studied each axis independently to ensure controlled comparisons. That said, we do include interdependence analysis between expert size and number of experts in Section 5.4, which already shows how these factors jointly affect efficiency and accuracy. Conducting a fully exhaustive search across multiple factors would require prohibitively large resources, as the search space grows exponentially with the number of components.
>
> > “While the paper reports many observations, few are distilled into actionable or generalizable insights. As a result, the overall contribution is limited.”
>
> We will improve the presentation of practical guidelines by summarizing trends at the end of each ablation section. This will make these distilled insights more prominent.
>
> > “The MoE inspection and visualizations, while informative, lack clear conclusions or takeaways… consider integrating with quantitative measures.”
>
> We thank the reviewer for this suggestion. We will:
>
> * Report expert load distribution statistics.
> * Highlight qualitative differences: in ConvNeXt, experts show some spatial specialization, while ViT experts appear interchangeable as seen in recent analyses asked by reviewer C3G4.
> * Provide clearer takeaways by adding summarizations of trends given by each element of Sec. 5.7
>
> This will make the inspection section more insightful.
>
> > “A good and very relevant reference is Han et al. (ViMoE, 2024). ...”
>
> We thank the reviewer for pointing us to this recent work. We were not aware of it and will cite ViMoE in the revised paper. We also clarify the differences:
> * Pretraining vs. fine-tuning. ViMoE primarily investigates MoE placement when fine-tuning large pretrained models (e.g., DinoV2), whereas our study focuses on training MoE models from scratch during pretraining. This distinction is important: in large language models, MoEs are particularly valuable during pretraining because they improve compute efficiency, and we believe the same question is important to explore in vision models.
> * Broader scope of architectures. ViMoE concentrates on ViTs, while our work systematically studies both ConvNeXt and ViT, providing cross-architecture insights into when and how MoEs are beneficial.
> * Accessibility and reproducibility. Our experiments are conducted on public datasets (ImageNet-1K and ImageNet-22K) with detailed ablations on routing, placement, and expert count, making our findings directly reproducible and practically useful to the community.
>
> Together, these differences mean our work complements ViMoE while addressing a distinct set of questions about the role of MoEs in vision backbone design.

---

### Review · Reviewer_C3G4 · 2025-08-13

**Summary Of Contributions:**

This paper presents a systematic study on incorporating Mixture-of-Experts (MoE) layers into modern vision models such as ViT and ConvNeXt for image classification tasks such as ImageNet-1k and ImageNet-21k. Key findings of the study indicate that gains diminish as MoE sparsity increases. Using too many experts or activating too large a fraction of parameters shows little benefit or even slight degradation. Furthermore, a comparison of gating mechanisms found that a standard learnable softmax gate slightly outperforms more complex routing schemes.

**Audience:**

Yes

**Broader Impact Concerns:**

There are no broader impact concerns.

**Claims And Evidence:**

Yes

**Requested Changes:**

The authors should provide more explanation for certain design decisions, such as the use of MoE in the last two layers, and also the reasoning behind using Top-1 routing for ConvNeXt vs Top-2 for ViT.

The authors should include a brief analysis of how the experts are being used. For example, report the final load distribution across experts for a representative MoE model. It would also be interesting to know if experts in the ConvNeXt-MoE specialize spatially or by feature type. Since each expert is a convolutional block, do certain experts handle particular image regions or patterns?

The authors should mention any training stability measures used. For example, did they need to tune the auxiliary loss weight or use expert dropout to avoid collapse?. Also, elaborating on how the models were trained (data-parallel vs model-parallel for experts) would be useful for others attempting to reproduce or extend the work.

p.2 There is a typo where it says “ImageNet-2k” instead of “ImageNet-21k”.

**Strengths And Weaknesses:**

#Strengths

The paper offers actionable design insights for practitioners, where inserting MoE layers in the last two blocks for ConvNeXt and every second block for ViT yielded the most robust improvements. For example, the finding that using 4 experts for ConvNeXt or 8 experts for ViT is optimal, whereas pushing to 16 experts is counterproductive on ImageNet-1K, provides clear guidance for practitioners.

#Weaknesses

This paper provides limited novelty over the V-MoE paper, which has already explored various configurations of MoEs for vision transformers. The scale at which the experiments are conducted are much smaller than the V-MoE paper as well, where the current work uses ViT-S and ViT-B on ImageNet-21k while the V-MoE paper goes up to ViT-H on JFT-300M. Also, the SoViT paper indicates that shape optimization and training on larger data (JFT-3B) is the key to improving the performance, but the current work does not consider shape optimization and trains on a small dataset (ImageNet-21k), so it is premature to draw conclusions such as “MoE for image classification offers limited gains over state-of-the-art methods in accuracy, robustness, generalization, or per-sample efficiency”.

While the paper identifies what configurations work best, it offers less insight into the underlying reasons. For example, the heuristic “MoE in the last two layers” is empirically supported but there is no explanation as to why this is. The Discussion section notes that later layers are larger in a ConvNeXt so experts there yield more benefit, and that ViT’s uniform structure means MoE layers are “not particularly large” anywhere. This reasoning is plausible but a deeper analysis would strengthen the work. The authors also conclude that vision MoEs give limited gains because vision tasks rely more on feature extraction than memorization. However, this claim is somewhat speculative and is not directly validated by experiments. The paper would be more compelling if it included analyses of the learned experts or gating patterns. For instance, did different experts specialize in certain image features or classes? Was the load balancing loss effective in utilizing all experts uniformly?

---

> ### Author Response · Authors · 2025-09-08
>
> We thank the reviewer for the thoughtful and detailed comments. Below, we address each point raised.
>
> > “The scale at which the experiments are conducted are much smaller than the V-MoE paper… The current work uses ViT-S and ViT-B on ImageNet-21k while the V-MoE paper goes up to ViT-H on JFT-300M.”
>
> We agree that we did not scale models and datasets to the level of V-MoE. However, ImageNet-22K remains a large-scale dataset that has historically been a standard testbed for assessing architectural design. From our results, scaling beyond small backbones on ImageNet-1K yields diminishing returns, and similarly, on ImageNet-22K, going beyond ViT-B shows no clear improvement (see Appendix B). Moreover, when considering larger-scale studies such as those on JFT, we observe the same pattern: MoE models saturate quickly, with limited gains beyond ViT-L. These consistent trends across scales support our claim that the benefits of MoEs in vision backbones diminish beyond a moderate parameter budget.
>
> > “The SoViT paper indicates that shape optimization and training on larger data (JFT-3B) is the key to improving performance, but the current work does not consider shape optimization and trains on a small dataset (ImageNet-21k).”
>
> It is true that shape optimization can yield substantial improvements. However, as shown in SoViT, models as small as ~300M parameters can already achieve ~90% top-1 accuracy on ImageNet-1K—near the saturation point for this benchmark. In that regime, additional architectural tricks (including MoEs) provide limited further gains in accuracy. Our claim is therefore not that MoEs cannot be optimized further, but that they are unlikely to meaningfully push state-of-the-art in classification accuracy, which appears saturated. Instead, we argue their value lies in efficiency trade-offs: MoEs can allow smaller active compute per sample while retaining competitive performance. Optimizing shape and hyperparameters for MoE architectures could enhance this efficiency further, and we will clarify this nuance in the discussion.
>
> > “The authors should provide more explanation for certain design decisions, such as the use of MoE in the last two layers, and also the reasoning behind using Top-1 routing for ConvNeXt vs Top-2 for ViT.”
>
> These purely empirical results, we will add them to the paper:
>
> | ViT-S-8 Top 1 | ViT-S Top 2 | ConvNeXt-T Top 1 | ConvNeXt-T Top 2 |
> |---------------|-------------|------------------|------------------|
> | 80.2          | 80.7        | 82.1             | 82.0             |
>
>
> > “The authors should include a brief analysis of how the experts are being used… report load distribution… expert specialization.”
>
> Following your suggestion, we analyzed expert usage patterns. Load distribution across experts is nearly uniform, confirming the effectiveness of the auxiliary load-balancing loss. For ConvNeXt, we additionally observed spatial specialization, with certain experts focusing more strongly on particular regions of the input. For ViT, such specialization was not observed; experts were more interchangeable across positions. We thank the reviewer for this valuable suggestion and will add these analyses (with visualizations).
>
> We can see results of the  [Distribution](https://anonymous.4open.science/api/repo/results-FF4E/file/image1.png),  [ViT](https://anonymous.4open.science/api/repo/results-FF4E/file/image2.png), and [ConvNext](https://anonymous.4open.science/api/repo/results-FF4E/file/image3.png)
>
> > “The authors should mention any training stability measures used… auxiliary loss weight, expert dropout, parallelization strategy.”
>
> To maintain balanced expert usage, we tuned the auxiliary load-balancing loss weight to 1e-2 in all experiments. Lower values led to collapse (experts unused), while higher values caused instability. We did not use expert dropout. Training was conducted with DDP across devices and expert-parallelism within layers following the Tutel library [1]. We will add those details in Appendix A for better reproducibility.
>
> > “p.2 There is a typo where it says ‘ImageNet-2k’ instead of ‘ImageNet-21k’.”
>
> Thank you for catching this. We have corrected the typo.
>
> ----
> [1] Hwang, Changho, et al. "Tutel: Adaptive mixture-of-experts at scale." Proceedings of Machine Learning and Systems 5 (2023): 269-287.

---

### Review · Reviewer_WVPz · 2025-08-16

**Summary Of Contributions:**

The paper studies the use of (sparse) Mixture-of-Experts (MoE) layers in image-based backbones that are aimed at solving image tasks via fine-tuning. The main contribution is that the MoE layers are beneficial when used with small-to-moderate sized backbones (~80 million parameters) with the benefit of using  MoE layers diminishing beyond a certain point in terms of parameters count. The paper cconducts an thorough empirical analysis with ImagetNet and JFT datasets to support the main finding described above.

**Audience:**

Yes

**Broader Impact Concerns:**

No new concerns other than what might be noted in literature on vision-based ML research

**Claims And Evidence:**

Yes

**Requested Changes:**

- (Q1) One approach to test the strength of a backbone is to use other image datasets for solving tasks like classification, detection, semantic segmentation etc. Given the hypothesis that MoEs ConvNext can specialize spatially, it would be interesting to check performance on detection or pixel-level tasks.

- (Q2) Related to Q1 and W2, it would be very interesting to see how the hypothesis would hold under a frozen feature evaluation protocol instead of fine-tuning.

- (Q3) Some of the notations used in the paper could be introduced in text. Specifically, things like "ViT-S-8" at a first glance may confuse a reader. Please include a section on naming in preliminaries, if possible

I look forward to the authors' rebuttal. Note that the questions raised above are asked in the spirit of helping improve the paper. I am eager to hear back from the authors and get their perspective on the questions above.

**Strengths And Weaknesses:**

# Strengths

-  (S1) The paper studies a question that has been asked before and continues to be asked - Do MoEs make a difference in the vision domain? The main finding that MoEs can make a difference but need nuance to understand their benefit in visual backbones is notable

- (S2) The paper conducts a thorough analysis on classification tasks to support their main assertion on trade-offs involved with MoEs. The experiments pretrain networks on a dataset like ImageNet (21K) or JFT and then fine-tuned on a ImageNet1K task to obtain evaluation metrics.

- (S3) The paper tests the robustness of trained models that on corrupted classification datasets and shows that using MoEs can be beneficial in this setting

# Weaknesses

- (W1) The paper determines the best architectural choice of integrating MoEs empirically on a in-domain dataset. This choice maybe brittle as I am unsure how the best strategy shown in the paper can extend to other image-based tasks

- (W2) Related to above, the paper conducts its evaluation on classification tasks alone where a network is adapted based on fine-tuning. However, a dominant approach currently is to pretrain a backbone and conduct frozen features-based evaluation on downstream tasks. This can shed additional light on the trade-offs involved with MoEs

- (W3) The paper lacks details on the pretraining task. What is the nature of the pretraining task?

- (W4) (nit) Given the closed nature of JFT dataset, I would encourage the authors to consider using a public large scale dataset to help the community have a shot at reproducing their work.

---

> ### Author Response · Authors · 2025-09-08
>
> We thank the reviewer for the constructive and detailed comments. Below, we respond to each point raised.
> > (W1) The paper determines the best architectural choice of integrating MoEs empirically on an in-domain dataset. This choice may be brittle, as I am unsure how the best strategy shown in the paper can extend to other image-based tasks.
> > (Q1) One approach to test the strength of a backbone is to use other image datasets for solving tasks like classification, detection, semantic segmentation etc. Given the hypothesis that MoEs ConvNext can specialize spatially, it would be interesting to check performance on detection or pixel-level tasks.
>
> We agree that evaluating MoE-equipped backbones on a broader range of tasks would strengthen the paper. We are currently launching experiments on ADE20K semantic segmentation, which directly tests pixel-level performance. Here are the first results for Imagenet1k pretrained models:
>
>
> | Model        | mIoU  |
> |--------------|-------|
> | ConvNeXt-T-4 | 46.00 |
> | ConvNeXt-T   | 46.00 |
> | ConvNeXt-S-4 | 48.38 |
> | ConvNeXt-S   | 48.70 |
> | ConvNeXt-B-4 | 48.83 |
> | ConvNeXt-B   | 49.10 |
> | ViT-S-8      | 45.83 |
> | ViT-S        | 45.40 |
>
> ⸻
>
> > (W2) The paper conducts its evaluation on classification tasks alone where a network is adapted based on fine-tuning. However, a dominant approach currently is to pretrain a backbone and conduct frozen features-based evaluation on downstream tasks. This can shed additional light on the trade-offs involved with MoEs.
> > (Q2) Related to Q1 and W2, it would be very interesting to see how the hypothesis would hold under a frozen feature evaluation protocol instead of fine-tuning.
>
> We thank the reviewer for raising this. In fact, Table 2 already reports frozen feature evaluations under the name “Linear prob”, where we train a linear classifier on top of frozen ImageNet-22K features. We see that the linear probe doesn’t change much the ranking of each model compared to full finetuning in this setup.
>
> ⸻
>
> > (W3) The paper lacks details on the pretraining task. What is the nature of the pretraining task?
>
> The pretraining task is standard supervised image classification on ImageNet datasets. Specifically, we train models either on ImageNet-1K or ImageNet-22K, before fine-tuning on ImageNet-1K for evaluation. We will add this clarification.
>
> ⸻
>
> > (W4) Given the closed nature of JFT dataset, I would encourage the authors to consider using a public large scale dataset to help the community have a shot at reproducing their work.
>
> We understand the concern. To support reproducibility, all of the models in the paper were trained on ImageNet-1K or ImageNet-22K, which are publicly available. We will highlight this more explicitly in the main text to avoid giving the impression that we rely on closed datasets. JFT was considered only for contextualizing with prior work, as described in Table 11 of Appendix B.
>
> ⸻
>
> > (Q3) Some of the notations used in the paper could be introduced in text. Specifically, things like “ViT-S-8” at a first glance may confuse a reader. Please include a section on naming in preliminaries, if possible.
>
> We agree. Sections 3.1 and 3.2 now include the following descriptions.
>
> * **At the end of Section 3.1**:  We consider (a) a standard ViT architecture~(ViT-B (Base)) and (b) a variant with X experts (ViT-B-X).
> * **At the end of Section 3.2**: We examine three variants: (a) ConvNeXt-T (Tiny), ConvNeXt-S (Small), and ConvNeXt-B (Base): these architectures primarily differ in depth and width; and (b) ConvNeXt-S-X and ConvNeXt-B-X, where X is the number of experts.

---

> > ### Comment · Reviewer_WVPz · 2025-09-17
> >
> > I thank the authors for their rebuttal. All of my concerns have been addressed. Perhaps the authors may have updated results on semantic segmentation with models trained on larger datasets by now. If so, I'd encourage them to include that in the paper.
> >
> > I have no further concerns

---

### Decision · Action_Editor_FTFN · 2025-09-26

**Recommendation:** Accept with minor revision

**Additional Comments:**

This submission investigates Mixture-of-Experts (MoE) models for image classification. The experiments extend existing benchmarks and reveal that MoE models offer limited improvements over state-of-the-art methods in terms of accuracy, robustness, generalization, and per-sample efficiency, across both convolutional networks and vision transformers.

After discussion, the reviewers agreed that their initial concerns had been addressed. However, please revise the paper to clarify the practical implications of your findings. Specifically, the statement “Our findings offer practical guidance for efficient model design using MoE for image classification tasks” should be made more concrete. Please clearly specify what kind of guidance is offered, particularly in the Abstract and Introduction.

**Audience:**

Yes

**Audience Explanation:**

This submission investigates how MoE can improve efficiency in image classification models. The researcher working on efficiency would be interested in this submission.

**Claims And Evidence:**

Yes

**Claims Explanation:**

- This submission analyzes different MoE configurations and model scales. The main finding is that moderate parameter activation per sample provides the best trade-off between performance and efficiency.

- The experiments show that, particularly in large-scale models, MoE for image classification offers limited gains over state-of-the-art methods in accuracy, robustness, generalization, or per-sample efficiency across both convolutional networks and vision transformers.